# Enhanced Recovery after Surgery (ERAS) Program for Patients with Peritoneal Surface Malignancies Undergoing Cytoreductive Surgery with or without HIPEC: A Systematic Review and a Meta-Analysis

**DOI:** 10.3390/cancers15030570

**Published:** 2023-01-17

**Authors:** Manuela Robella, Marco Tonello, Paola Berchialla, Veronica Sciannameo, Alba Maria Ilari Civit, Antonio Sommariva, Cinzia Sassaroli, Andrea Di Giorgio, Roberta Gelmini, Valentina Ghirardi, Franco Roviello, Fabio Carboni, Piero Vincenzo Lippolis, Shigeki Kusamura, Marco Vaira

**Affiliations:** 1Unit of Surgical Oncology, Candiolo Cancer Institute, FPO-IRCCS, 10060 Torino, Italy; 2Advanced Surgical Oncology Unit, Surgical Oncology of the Esophagus and Digestive Tract, Veneto Institute of Oncology IOV-IRCCS, 35128 Padova, Italy; 3Center for Biostatistics, Epidemiology and Public Health (C-BEPH), Deptartment of Clinical and Biological Sciences, University of Torino, 10124 Torino, Italy; 4Abdominal Oncology Department, Fondazione Giovanni Pascale, IRCCS, 80131 Naples, Italy; 5Surgical Unit of Peritoneum and Retroperitoneum, Fondazione Policlinico Universitario A. Gemelli-IRCCS, 00168 Rome, Italy; 6SC Chirurgia Generale d’Urgenza ed Oncologica, AOU Policlinico di Modena, 41125 Modena, Italy; 7UOC Ovarian Carcinoma Fondazione Policlinico Universitario “A. Gemelli” IRCCS, Università Cattolica del Sacro Cuore, 00168 Rome, Italy; 8Unit of General Surgery and Surgical Oncology, Department of Medicine, Surgery, and Neurosciences, University of Siena, 53100 Siena, Italy; 9Peritoneal Tumours Unit, IRCCS Regina Elena National Cancer Institute, 00144 Rome, Italy; 10General and Peritoneal Surgery, Hospital University Pisa, 56124 Pisa, Italy; 11Peritoneal Surface Malignancies Unit, Fondazione Istituto Nazionale Tumori IRCCS Milano, 20133 Milano, Italy

**Keywords:** ERAS, peritoneal cancer, cytoreductive surgery, perioperative management, chemotherapy, HIPEC

## Abstract

**Simple Summary:**

Cytoreductive surgery and Hyperthermic IntraPEritoneal Chemotherapy (HIPEC) represent a promising treatment strategy for the management of selected cases of peritoneal cancer, but it’s still burdened by significant morbidity and prolonged hospital stay. Herein, the review of the impact of ERAS program on length of stay, postoperative complications and readmission rate in patients undergoing cytoreductive surgery with or without HIPEC for peritoneal surface malignancies.

**Abstract:**

Enhanced recovery after surgery (ERAS) program refers to a multimodal intervention to reduce the length of stay and postoperative complications; it has been effective in different kinds of major surgery including colorectal, gynaecologic and gastric cancer surgery. Its impact in terms of safety and efficacy in the treatment of peritoneal surface malignancies is still unclear. A systematic review and a meta-analysis were conducted to evaluate the effect of ERAS after cytoreductive surgery with or without HIPEC for peritoneal metastases. MEDLINE, PubMed, EMBASE, Google Scholar and Cochrane Database were searched from January 2010 and December 2021. Single and double-cohort studies about ERAS application in the treatment of peritoneal cancer were considered. Outcomes included the postoperative length of stay (LOS), postoperative morbidity and mortality rates and the early readmission rate. Twenty-four studies involving 5131 patients were considered, 7 about ERAS in cytoreductive surgery (CRS) + HIPEC and 17 about cytoreductive alone; the case histories of two Italian referral centers in the management of peritoneal cancer were included. ERAS adoption reduced the LOS (−3.17, 95% CrI −4.68 to −1.69 in CRS + HIPEC and −1.65, 95% CrI −2.32 to –1.06 in CRS alone in the meta-analysis including 6 and 17 studies respectively. Non negligible lower postoperative morbidity was also in the meta-analysis including the case histories of two Italian referral centers. Implementation of an ERAS protocol may reduce LOS, postoperative complications after CRS with or without HIPEC compared to conventional recovery.

## 1. Introduction

Peritoneal surface malignancies (PSM) are both a consequence of different primary tumors, synchronous or metachronous, and the clinical presentation of primitive peritoneal neoplasms. Despite significant recent advances in the management of peritoneal metastases, this diagnosis still is frequently linked to a poor prognosis. The unfavourable outcome is often accompanied by clinical symptoms that dramatically impact on quality of life and represent a real challenge for the managing health care provider.

Cytoreductive surgery (CRS) eventually associated with hyperthermic intraperitoneal chemotherapy (HIPEC) has emerged as a promising therapeutic option for highly selected patients with PSM.

The cytoreductive surgical approach focuses on the removal of the primary tumor, of the peritoneal area interested by the disease (peritonectomy) often associated with multivisceral resection (omentectomy, oophorectomy, bowel resection, spleen and/or gallbladder removal) in order to obtain no residual disease. Following CRS, a peritoneal lavage with chemotherapeutic agents is typically performed within the abdominal cavity for 30–90 min at a temperature of 41–42 °C to treat remaining microscopic peritoneal disease, achieving high peritoneal concentrations with limited systemic absorption [1].

Despite its efficacy, CRS-HIPEC is a complex and technically challenging procedure with potential high morbidity and mortality [2,3], which makes patient selection and institutional experience essential for optimal treatment and prevention of adverse events [4]. Moreover, postoperative complications after CRS-HIPEC are independent prognostic factors on survival [5].

Enhanced recovery after surgery (ERAS) is an evidence-based multimodal approach developed in order to facilitate earlier recovery after surgery and accelerate postoperative rehabilitation. The key elements of ERAS program include patient and family education and counseling, patient optimization prior to admission, minimal fasting (light meal up to six hours before surgery, carbohydrate beverage two hours before anesthesia), multimodal analgesia with appropriate use of opioids, nausea and vomiting prophylaxis, early nutrition and mobilization [6].

ERAS program has been applied to numerous surgical fields, first and foremost colorectal cancer surgery, reporting a decrease in postoperative complication rates, a shortening in hospital lengths of stay and consequently, a costs reduction. Furthermore, in the last decades ERAS protocol obtained similar benefits in other multiple types of digestive and major surgical procedures [7,8]. Despite this worldwide diffusion and the release of ERAS Society Guidelines specific to CRS with or without HIPEC, the adoption in this field is still disappointingly low. On the basis of these considerations an updated specialistic review of the literature is needed to assess the real applicability, safety and efficacy of ERAS in peritoneal surface malignancies management.

The goal of this meta-analysis was to investigate the impact of ERAS program on LOS, readmission rate, postoperative complications and reoperation rate evaluating the influence of ERAS elements on postoperative recovery and the compliance to this pathway.

## 2. Materials and Methods

### 2.1. Eligibility Criteria and Study Selection

Inclusion criteria: retrospective and prospective cohort studies, case-control and randomized control studies comparing ERAS program adoption with standard perioperative care for CRS associated or not with HIPEC for peritoneal surface malignancies of different origin were considered for inclusion. The unpublished experience of two Italian institutes that currently apply the ERAS protocol in this setting were included in the analysis.

Exclusion criteria: we excluded abstracts, letters to editor, study protocol, non-English language papers, case reports. Studies which enrolled a population of patients undergoing gynaecological surgery solely for benign indications or for basic pelvic surgery were excluded.

### 2.2. Data Source and Extraction

Literature search, study design, data extraction and analysis were performed according to the preferred reporting items for systematic reviews and meta-analyses (PRISMA) statements [9] (Figure 1).

The search for scientific papers contained the following combinations of keywords “hyperthermic intraperitoneal chemotherapy” or “HIPEC” and “enhanced recovery after surgery” or “ERAS” and “cytoreductive surgery” or “peritoneal carcinomatosis” and “ERAS” or “enhanced recovery after surgery” without any language filter. The search for articles was carried out using the following databases: MEDLINE, PubMed, EMBASE, Google Scholar and Cochrane Database.

The search was limited to studies published from inception to December 2021.

Only reports on ERAS + CRS or ERAS + HIPEC + CRS were retained. All studies of interest were obtained as full-text articles. All publications related to ERAS and CRS + HIPEC, including clinical reports and systematic and narrative reviews, were considered to retrieve the maximum number of publications.

The abstracts of the selected papers were analysed to identify those meeting the inclusion criteria. Papers that did not have any of the study outcomes, did not address ERAS protocol, did not undergo CRS with or without HIPEC, were excluded.

The references for all included papers, review articles, commentaries, and editorials on this topic were also reviewed to identify other studies of interest that were missed during the primary search. Pertinent references and electronic links were hand-searched, and cross-referencing was done for selected articles.

Outcome of interest were LOS, readmission rate, complication rate according to Clavien-Dindo classification, reoperation rates and mortality.

The data extraction was carried out independently by two study investigators. All the articles were collected to extract the most pertinent information from the studies, for instance publication year, study type, sample size, mean age, gender, peritoneal cancer index (PCI) or Aletti score, primary tumor, HIPEC drugs, mean operative time and ERAS protocol application.

Quality of the studies was assessed using the risk of bias in non-randomized intervention tool ROBINS-I. Publication bias was assess using Egger’s test. The systematic review followed the recommendations of the Preferred Reporting Items for Systematic Reviews and Meta-Analyses (PRISMA). The protocol has not been registered.

### 2.3. Statistical Analysis

In the main analysis, for each outcome we performed an arm-based Bayesian Meta-Analysis (BMA) [10], which uses data from each treatment arm to describe population-averaged effect size, allowing to include both two and single-arm studies. Studies that did not report the outcome were excluded from the meta-analysis.

We applied a hierarchical Bayesian model assuming heterogeneity of variances of the random effects. We assigned an inverse Wishart prior to the unstructured variance-covariance matrix. For binary outcomes we selected a logit link function. We set the shape and scale parameters of inverse gamma priors for variance of random effects both to 0.001, which is the default value. Then, we constructed three Markov Chain Monte Carlo (MCMC) chains, and we set the number of iterations for adaptation in MCMC algorithm to 5000 (the default value). We run 200,000 iterations in each MCMC, and we used 100,000 iterations as burn-in period and we set the thinning rate to 1. We used the Gelman-Rubin convergence diagnostics to assess the convergence of the MCMC models, which was considered adequate if a value less than 1.05 was reached [11].

Then, we conducted a sensitivity analysis through an Individual Bayesian Meta-Analysis (IBMA), which allows analyzing both two-arms aggregated data and two-arms individual-level data. The prior distributions for the intercept and for the treatment effects were normal with a scale parameter of 10. The prior distribution of heterogeneity was half-normal, with a scale of 2.5. We constructed four MCMC chains, and as in the main analysis, we set the number of iterations for adaptation in MCMC algorithm to 5000, we ran 200,000 iterations in each chain, and we used 100,000 iterations as burn-in, with a thinning rate of 1. For both the analyses, the treatment effects of continuous outcomes were expressed in terms of median difference values and 95% credibility intervals (95% CrI) of the posterior distributions. For binary outcomes, Odds Ratios (OR) were expressed in terms of median values and 95% CrI of the posterior distributions.

To help with interpretation, we reported the probability of direction (pd), which is the probability that an effect goes in a particular direction [12]. So, for example, if the estimated OR is <1, the pd is the proportion of the posterior distribution with values <1. The pd is strongly correlated with the *p*-value and can be used to draw parallels and give some reference to readers non-familiar with Bayesian statistics [13]. The threshold beyond which the effect is considered as non-negligible is 0.975.

We used the R software version 4.1.2 [14] and more in detail the R packages *pcnetmeta* [15] for the BMA and *multinma* [16] for the IBMA.

## 3. Results

### 3.1. Study Characteristics

A total of 8391 articles, reports and clinical studies on cytoreductive surgery with or without HIPEC were identified. After removing duplicates, 6891 abstracts were evaluated and screened for eligibility: 24 studies involving 5131 patients were included in the final analysis of the systematic review and meta-analysis: 7 were selected for CRS + HIPEC and 17 for CRS alone.

We must notice that the studies concerning CRS alone turned out to be focused purely on gynaecological surgery: we must underline that they have been carefully selected in order to include only papers concerning CRS for peritoneal cancer, excluding those relating to simple procedures such as annessiectomy and/or hysterectomy.

In the CRS + HIPEC group a total of 743 patients were included, with 434 treated according to the ERAS pathway and 309 with standard recovery. Six studies were case-control and only one was a retrospective single cohort analysis.

In the CRS group, all the included studies compared patients treated according to the ERAS program to standard management. In the study published by Kalogera [17], three populations were considered separately: cytoreduction, staging, and vaginal surgery: for consistency with the inclusion criteria of our study, only patients submitted to CRS were included in our analysis.

The detailed characteristics of the populations and surgery of the included papers are reported in Table 1 and Table 2.

Data regarding LOS, postoperative complications and readmission rate are summarized in Table 3.

### 3.2. Postoperative Length of Stay

Six studies in the CRS + HIPEC group (*n* = 587) and 17 studies in the CRS group (*n* = 4388) reported postoperative LOS. Compared to standard recovery, patients included in the ERAS pathway had shorter LOS by −3.17 days (95% CrI −4.68 to −1.69) and −1.65 days (95% CrI −2.32 to −1.06), respectively. Probability of direction (pd) smaller than 0.95 did not show a strong association between ERAS and LOS in IBMA (Table 4).

### 3.3. Postoperative Morbidity and Mortality Rate

IBMA showed a non-negligible reduction in postoperative major complications in both CRS + HIPEC (OR = 0.48, 95% CrI 0.22, 0.98 with pd = 0.98) and CRS alone (OR = 0.58, 95% CrI 0.32, 0.94 with pd = 0.98). Similarly, a non-negligible reduction in complication is observed in the IBMA for both CRS + HIPEC with 1 study with aggregated level data and the 2 IPD studies (OR = 0.31, 95% CrI 0.10, 0.89 with pd = 0.99) and CRS alone with 10 studies with aggregated level data and the 2 IPD studies (OR = 0.56, 95% CrI 0.44, 0.71 with pd = 0.99).

No strong association was observed between ERAS and reduction of reoperation rates in 3 papers about CRS + HIPEC and 6 about CRS alone.

Only two studies about HIPEC and 10 studies about CRS alone evaluated the postoperative mortality: negligible difference between the two groups was detected, even considering two-arms individual level data (Table 4).

### 3.4. Readmission Rate

Data on readmission were reported by 4 studies about HIPEC and by all the studies about CRS alone: pooled analysis demonstrated negligible difference in the risk of early readmission) in both groups treated with ERAS program (Table 4).

The aforementioned data are summarized in Table 4.

The risk of bias for each study is reported in Figure 2 and showed not particular concerns, ranging from low to moderate in the majority of the studies. For those studies presenting serious concerns, the major issue identified was related to confounding due to not adjusted statistical analysis.

## 4. Discussion

The key principle of the ERAS protocol is to standardize and optimize perioperative patients care in order to reduce the bodily stress reactions caused by injury, associated with adverse outcomes. The program includes pre-operative counselling and nutritional screening, avoidance of perioperative fasting and carbohydrate loading up to 2 h preoperatively, standardized anaesthetic and analgesic regimens (epidural and non-opioid analgesia), controlled perioperative fluid management, early feeding and mobilization.

The effects of ERAS have been extensively investigated in standard surgical settings, including colorectal and gynecological surgery [41,42]. Its application is slowly spreading even to more complex and particular interventions such as those for peritoneal metastases [17,18,19,20,21,22,23,24,25,26,27,28,29,30,31,32,33,35,36,37,38,39,40] and guidelines for perioperative care in CRS with or without HIPEC have recently been published by the ERAS Society detailing the benefits of each item of the pathway [43,44].

This is the first meta-analysis concerning the application of the ERAS program focused on patients suffering from peritoneal disease of various origins and subjected to heterogeneous surgical procedures. Moreover, it differs from previous reviews in its rigorous evaluation of the included studies considering only populations submitted to cytoreductive surgery with or without HIPEC for peritoneal malignancies. Considering that reporting of ERAS for CRS and especially for CRS and HIPEC are minimal, we also included the unpublished experiences of two referral centers in the treatment of PSM.

In this meta-analysis the application of the ERAS program was associated with a significant reduction in the LOS and lower postoperative morbidity and mortality rates compared to the standard management.

This analysis confirmed the results obtained in other types of surgery including colorectal, upper gastrointestinal or gynecologic cancer. Furthermore, these data agreed with the results of two previous smaller meta-analyses that separately evaluated ERAS in CRS and in CRS + HIPEC, demonstrating a reduction of LOS, complications and costs without increasing rates of reoperation and mortality [42,45].

As already reported in previous studies, this meta-analysis revealed the heterogeneous components of the ERAS protocol and its different application and implementations across trials [46,47] (Table 5).

Full compliance is difficult to reach, especially considering the heterogeneity of pathology and surgical procedures. We should moreover consider that surgical management of peritoneal surface malignancies with CRS +/− HIPEC is an aggressive approach, often requiring multiple visceral resections in patients with advanced disease submitted in most cases to several lines of systemic chemotherapy. This procedure is characterized by long periods of extreme surface exposure, which may result in a significant loss of fluids and proteins and a decrease of the intravascular volume; these effects can be even more evident if the procedure is followed by perfusion with intraperitoneal chemotherapy (HIPEC) at 42 °C. In this setting ERAS items are sometimes not practical: for example, patients submitted to an extended gastric surgery necessitate the use of NGT and take longer to return to oral nutrition. As well, in some cases it’s not possible to perform epidural anaesthesia. These necessary deviations from the ERAS program partly justify the heterogeneity of compliance to the protocol across the different studies. While compliance to preoperative and intraoperative items is high, the adhesion to intraoperative and postoperative recommendations decreases [48].

Preoperative counseling, nutritional supplement, avoidance of bowel preparation and carbohydrate loading are ERAS recommendations adopted in almost all studies.

Another essential component of ERAS program is multimodal pain management: poorly controlled postoperative pain may cause delay recovery and prolong the LOS: in this meta-analysis the multimodal analgesia opioid sparing is adopted in 22 out of 24 centers.

Although goal fluid therapy is one of the mainstays of the ERAS program [49,50,51], concerns about HIPEC-induced nephrotoxicity and the replacement of large-volume ascites led to a liberal fluid management. In some studies, the fluid restriction was associated with a higher percentage of major postoperative complications [52] while in other analyses it’s linked to a shorter LOS and lower postoperative morbidity rate without increasing the rate of acute kidney injury or renal dysfunction [53]. In spite of the fact that certain agents utilized for HIPEC as cisplatin are linked to a greater risk of renal injury, in this meta-analysis no difference in terms of postoperative complications are related to the chemotherapic drug used; moreover, in the study published by White [23], cisplatin administration was strongly associated with acute kidney injury before ERAS but not afterward. In this meta-analysis the goal-fluid therapy was adopted in 19 centers out of 24. Considering the peculiar surgical setting we cannot ignore the difficult applicability, at least in a standard manner, of this ERAS item: surgical times are heterogeneous (in any case longer than a standard colorectal surgery), procedures are in most cases performed through a laparotomic approach resulting in significant hydro-electrolytic loss; lastly, the thermal damage of the electro-evaporation of the peritoneum causes massive loss of oncotic proteins.

The routine replacement of abdominal drainages after surgery is still a hotly debated topic: in most cases, especially if HIPEC is performed, they are used to prevent the formation of intraabdominal collections after extensive and aggressive CRS [24].

In the postoperative management, the ERAS program recommends the early mobilization and the early introduction of oral feeding. Although it’s proved that the oral feeding is the best way to stimulate peristalsis, some concerns are due to the high rate of postoperative ileus secondary to the heated intraperitoneal chemotherapy and the impossibility to start early the oral nutrition if multiple gastrointestinal resection were performed. The mechanism by which ERAS program itself decreases the rate of postoperative ileus is multifactorial: drink clear fluids up to two hours prior to the procedure helps preventing dehydration before surgery, decreasing narcotic use reduces the effect on bowel motility; moreover, goal directed fluid therapy decreases ileus rates secondary to bowel edema [54]. In the pooled analysis 21 out of 24 Institutions adopted the resumption of feeding by mouth by the second postoperative day.

Although compliance with the program has been shown to be crucial to achieve optimal care for the surgical patients in different specialties [29,55,56,57,58], even when a patient does not fully comply with all the items (due to the heterogeneous surgical procedures) consistent benefits from the implementation of the standard management have been reported. A dose-response relationship between compliance and LOS and postoperative morbidity rate reduction has been described, while a poor compliance to ERAS elements is an independent predictor of early readmission [28,59,60].

Improvements in postoperative recovery may be especially meaningful in this oncological patient population: in fact, ERAS remained the strongest predictor of timely resumption of adjuvant systemic chemotherapy [36] because patients maintain their normal physiology postoperatively and recover faster from surgery. 

Strength of this meta-analysis is the heterogeneous population affected by peritoneal surface malignancies of different origins, submitted to various surgical procedures: this improves the applicability of the ERAS program and its generalizability even outside of gynaecological surgery.

Limitation of this meta-analysis is the lack of randomized controlled trials; almost all the studies are based on non-randomized historical cohorts, but we should think that randomized trials of ERAS would not be ethically feasible considering the growing breadth of evidence of the effectiveness of ERAS.

## 5. Conclusions

Due to the historic high morbidity and mortality associated with CRS especially if combined with HIPEC, surgeons are often hesitant to implement a full ERAS program compared to more conservative management.

This meta-analysis supported the idea that in selected patients affected by peritoneal surface malignancies submitted to CRS with or without HIPEC, the implementation of ERAS protocol is safe and feasible and may offer significant improvements in outcomes. The compliance to the program is a crucial element to obtain shorter postoperative hospitalization, reduce postoperative complications without increasing readmissions rates.

## Figures and Tables

**Figure 1 cancers-15-00570-f001:**
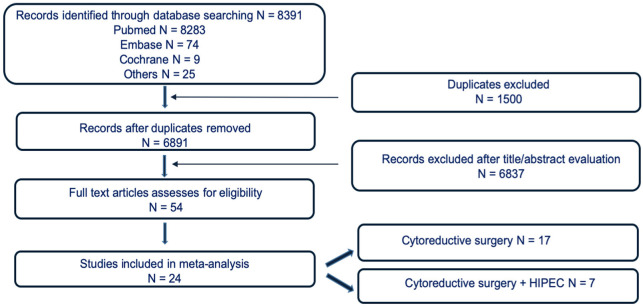
Study Flow Chart.

**Figure 2 cancers-15-00570-f002:**
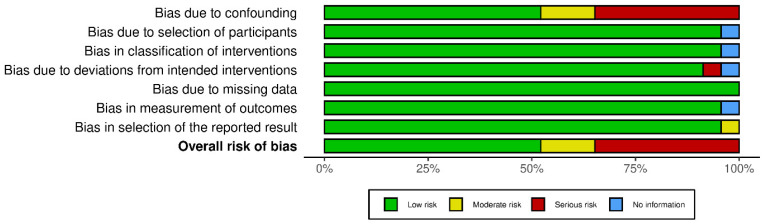
Risk of bias of the studies: traffic light plot.

**Table 1 cancers-15-00570-t001:** Details of included studies about CRS + HIPEC.

Study	Sample Size		Mean Age (Years)		Female (%)		PCI		Primary Tumor		HIPEC Drug		Mean Operative Time (min)	
	ERAS	Control	ERAS	Control	ERAS	Control	ERAS	Control	ERAS	Control	ERAS	Control	ERAS	Control
Duzgun 2019 [18]	62	40	57.3	56.1	58.1	57.5	12.8	12.4	CRC 26 OC 11 STS 8 GC 5 Other 12	CRC 20 OC 8 STS 6 GC 5 Other 11	NR	NR	472	395
Martin 2020 [19]	20	105	51.7 *	58.7 *	12	43	3 *	5.5 *	PMP 9 CRC 8 GC 1 PP 1 Other 1	PMP 7 CRC 4 GC 3 PP 7 Other 10	CDDP 1 IRI 2 MMC 11 OXA 6	CDDP/DXR 3 CDDP 2 IRI 1 MMC 55 OXA/IRI 11 OXA 27 Other 6	270 *	300 *
Siddhartan 2020 [20]	15	16	60 *	57 *	NR	NR	3 *	6 *	NR	NR	MMC	MMC	418 *	452 *
Webb 2020 [21]	81	49	54.4	56.0	39	43	12	11.5	PMP 47 CRC 18 MPM 7 OC 2 GC 5 Other 2	PMP 26 CRC 14 MPM 4 OC 3 GC 0 Other 2	CDDP MMC	CDDP MMC	390	390
Lu 2020 [22]	20	11	50	47	12	43	13.5	10	PMP 12 CRC 4 Other 4	PMP 7 CRC 4	MMC	MMC	347	391
White 2021 [23]	80	88	56.5	56.7	62.5%	63.6%	13.2	13.6	NR	NR	CDDP 23.8% MMC 76.2%	CDDP 26.1% MMC 73.9%	370	360
Cascales Campos 2016 [24]	156		57 *		148 (94.9%)		8 (0–32)		OC 113 CRC18 PMP 13 STS 5 Other 5		OC: Paclitaxel/CDDP PMP and CRC: MMC Sarcomas: CDDP + DXR		300 *	
Veneto Institute of Oncology	33		62.3		22 (66.6%)		15.4		OC 12 CRC 7 MPM 3 PMP 5 STS 2 Other 4		CDDP 10 MMC 1 CDDP + MMC 15 CDDP + DXR 7		558.2	
Candiolo Cancer Institute	28		58.57		18 (64.28%)		11,2		OC 4 CRC 9 MPM 1 PMP 13 PP 1		CDDP 7 MMC 3 CDDP + MMC 18		341.4	

CRC = Colorectal Cancer; OC = Ovarian Cancer; GC = Gastric Cancer; PMP = Pseudo Myxoma Peritonei; MPM = Malignant Peritoneal Mesothelioma; PP = Primary Peritoneal cancer; STS = Soft Tissue Sarcoma; CDDP = cisplatin; MMC = mytomicin C; OXA = oxaliplatin; DXR = doxorubicin; * = median.

**Table 2 cancers-15-00570-t002:** Details of included studies about CRS.

Study	Sample Size		Mean Age (Years)		Female (%)		Aletti Score		Primary Tumor		Operative Time (min)	
	ERAS	Control	ERAS	Control	ERAS	Control	ERAS	Control	ERAS	Control	ERAS	Control
Marx, 2006 [25]	69	72	61 *	62 *	69 (100%)	72 (100%)	NR	NR	OC 61	OC 62	134 *	122 *
Gerardi, 2008 [26]	19	45	62	58	19 (100%)	45 (100%)	NR	NR	OC 19	OC 45	NR	NR
Kalogera, 2013 [17]	81	78	64.3	65.1	81 (100%)	78 (100%)	NR	NR	Gyn 100%	Gyn 100%	227	278
Myriokefalitaki, 2016 [27]	99	99	61.6	61.0	99 (100%)	99 (100%)	High 46	High 58	Gyn 100%	Gyn 100%	NR	NR
Modesitt, 2016 [28]	136	211	51.8	51.1	136 (100%)	211 (100%)	NR	NR	Gyn 100%	Gyn 100%	228	222
Bisch, 2018 [29]	367	152	57 *	61 *	367 (100%)	152 (100%)	Low 253 Med/High 114	Low 94 Med/High 58	OC EC Benign	OC EC Benign	NR	NR
Agarwal, 2018 [30]	45	45	53 *	58 *	45 (100%)	45 (100%)	NR	NR	OC 45	OC 45	229	219
Boitano, 2018 [31]	179	197	55.9	57.8	179 (100%)	197 (100%)	Low 124 Mod 46 High 9	Low 144 Mod 47 High 6	OC 62 Uterine 20 Cervix 4 Others 93	OC 59 Uterine 40 Cervix 6 Others 92	NR	NR
Meyer, 2018 [32]	533	74	58	58	533 (100%)	74 (100%)	NR	NR	Adv. OC 288	Adv. OC 48	216 *	236 *
Bergstrom, 2018 [33]	109	158	55.2	51.7	109 (100%)	158 (100%)	NR	NR	Adv. OC 54	Adv. OC 41	285	238
Bernard, 2020 [34]	187	441	58.6	60.3	187 (100%)	441 (100%)	NR	NR	OC 129 Uterine 36 Cervix 22	OC 335 Uterine 101 Cervix 5	145 *	121 *
Sanchez-Iglesias, 2020 [35]	50	49	57.8	57.2	50 (100%)	49 (100%)	Low 11 Med 16 High 23	Low 6 Med 17 High 26	OC 48 Others 2	OC 47 Others 2	288	287
Tankou, 2021 [36]	128	150	NR	NR	128 (100%)	150 (100%)	Low 90 Med28 High 10	Low 114 Med 33 High 3	OC/PP 120 Uterine 8	OC/PP 150 Uterine 0	NR	NR
Ferrari, 2020 [37]	83	85	56.5	54.9	83 (100%)	85 (100%)	NR	NR	Adv. OC 24	Adv. OC 25	139	160
Mendivil 2018 [38]	86	91	63.87	56.01	86 (100%)	91 (100%)	NR	NR	Gyn 100%	Gyn 100%	NR	NR
Kay, 2020 [39]	94	42	63.1	60.1	94 (100%)	42 (100%)	NR	NR	OC 100%	OC 100%	NR	NR
Reuter, 2021 [40]	47	87	65 *	60 *	47 (100%)	87 (100%)	NR	NR	OC 100%	OC 100%	303 ± 91	306 ± 103
Veneto Institute of Oncology	33		66.87		30 (90.9%)		PCI 12.24		OC 24 CRC 2 PMP 1 Other 6		363.63	
Candiolo Cancer Institute	33		61		26 (78.7%)		PCI 12.5		OC 17 CRC10 PMP 6		244	

CRC = ColoRectal Cancer; OC = Ovarian Cancer; GC = Gastric Cancer; PMP = PseudoMyxoma Peritonei; MPM = Malignant Peritoneal Mesothelioma; PP = Primary Peritoneal cancer; * = median.

**Table 3 cancers-15-00570-t003:** Details of outcomes of the included studies.

	Study	Arms	Sample Size N		LOS Mean (Days)		LOS (SD)		Readmission N		Reoperation N		Complications N		Major complic. N		Death N	
			Control	ERAS	Control	ERAS	Control	ERAS	Control	ERAS	Control	ERAS	Control	ERAS	Control	ERAS	Control	ERAS
CRS + HIPEC	P.A Cascales Campos 2016 [24]	1		156						14				48		21		
	Duzgun 2019 [18]	2	40	62	10	7	4.50	1,10			2	2	14	9	4	1	3	3
	Webb 2020 [21]	2	49	81	10.30	6.90	8.90	5	7	13	6	5			12	12		
	Siddharthan 2020 [20]	2	16	15	11	7	2.38	16.30							4	3		
	Lu 2020 [22]	2	11	20	9	6	2.96	1.85	3	1					1	1		
	Martin 2020 [19]	2	105	20	11	9	28.96	16.30	19	6	7	1						
	White 2021 [23]	2	88	80	10	7.90	4.50	6.40	12	13					34	19	19	1
CRS	Marx 2006 [25]	2	72	69	6	5	45	21.50	7	2	9	3	23	17	18	4	2	0
	Gerardi 2008 [26]	2	45	19	11.40	8.70	18	17	15	4					26	12		
	Kalogera, 2013 [17]	2	78	81	10.70	6.50	11.40	3.50	14	21			56	51	16	17	1	1
	Myriokefalitaki, 2016 [27]	2	99	99	7.20	4.30	5.68	2.78	6	5			27	30	4	2	1	0
	Modesitt, 2016 [28]	2	211	136	3	2	0.74	0.74	13	7	1	2	85	29			2	0
	Bisch 2018 [29]	2	152	367	6.40	4.50	7.50	5.90	10	22			81	133			0	2
	Boitano 2018 [31]	2	197	179	4	2.90	2.40	1.90	21	18								
	Bergstrom 2018 [33]	2	158	109	5	5.50	2.96	2.96	15	13					32	15		
	Meyer 2018 [32]	2	74	533	4	3	6.75	14	10	70	4	12					0	1
	Mendivil 2018 [38]	2	91	86	8.04	4.88	7.19	4.23	4	2								
	Agarwal 2019 [30]	2	45	45	6	4	10.40	4.44	5	1			17	11				
	Sanchez-Iglesias 2020 [35]	2	49	50	9	7	3.70	2.96	10	3	5	4	35	30	8	9	2	2
	Ferrari 2020 [37]	2	85	83	7	4	5.18	2.22	5	4			28	15	8	1		
	Bernard 2020 [34]	2	441	187	4.70	3.80	3.80	3.20	35	9	7	5	107	30			2	0
	Kay, 2020 [39]	2	42	94	6.7	4.2			3	9								
	Tankou, 2021 [36]	2	150	128	4	3	1.48	0.74	13	14	2	0					1	0
	Reuter, 2021 [40]	2	87	47	13	11	3.7	2.22	18	7			46	14	9	2		

**Table 4 cancers-15-00570-t004:** Bayesian Meta-analyses on aggregated data (AD) and individual participant data (IPD) results.

Outcome	Treatment	BMA AD 2 Arms + AD 1 Arms Median (95% CrI)	pd	IBMA AD 2 Arms + IPD Median (95% CrI)	pd	No AD Studies with 2 Arms	No AD Studies with 1 Arm	No IPD Studies 2 Arms
Hospital stay (days)	HIPEC + ERAS vs. HIPEC	−3.17 (−4.68, −1.69)	0.99	−3.00 (−7.84, 1.55)	0.90	6	0	2
	CRS + ERAS vs. CRS	−1.65 (−2.32, −1.06)	0.99	−1.28 (−3.01, 0.39)	0.93	17	0	2
Major complications	HIPEC + ERAS vs. HIPEC	0.53 (0.18, 1.59)	0.88	0.48 (0.22, 0.98)	0.98	5	1	2
	CRS + ERAS vs. CRS	0.70 (0.33, 1.52)	0.83	0.58 (0.32, 0.94)	0.98	8	0	2
Reoperation	HIPEC + ERAS vs. HIPEC	0.63 (0.09, 4.48)	0.69	0.58 (0.15, 2.07)	0.82	3	0	2
	CRS + ERAS vs. CRS	0.65 (0.17, 2.54)	0.74	0.67 (0.27, 1.64)	0.83	6	0	2
Readmission	HIPEC + ERAS vs. HIPEC	0.84 (0.23, 2.84)	0.62	1.16 (0.63, 2.08)	0.69	4	1	2
	CRS + ERAS vs. CRS	0.79 (0.48, 1.28)	0.84	0.84 (0.64, 1.09)	0.92	17	0	2
Complications	HIPEC + ERAS vs. HIPEC	0.57 (0.05, 7.31)	0.70	0.31 (0.10, 0.89)	0.99	1	1	2
	CRS + ERAS vs. CRS	0.61 (0.37, 1.01)	0.95	0.56 (0.44, 0.71)	0.99	10	0	2
Death	HIPEC + ERAS vs. HIPEC	0.23 (0.02, 3.40)	0.87	0.15 (0.01, 1.76)	0.94	2	0	2
	CRS + ERAS vs. CRS	0.49 (0.08, 3.25)	0.79	0.43 (0.05, 1.96)	0.88	10	0	2

BMA = Bayesian Meta-Analysis; IBMA = Individual Bayesian Meta-Analysis; AD = Aggregated level Data; IPD = Individual Participant Data; 95% CrI = 95% Credible Interval; pd = Probability of Direction.

**Table 5 cancers-15-00570-t005:** ERAS elements in the included studies.

	Study	Preoperative Information/ Counseling	Nutritional Supplement	No Bowel Preparation	Carbohydrate Loading	Multimodal Analgesia	PONV Management	Goal Directed Fluid Therapy	Avoidance Abdominal Drains	Avoidance NGT	Early NGT Removal (<24 h)	Early UC Removal (<24 h)	Time to Fluid Intake (<24 h)	Early Solid Intake (<48 h)	Early Mobilization/ Deambulation
CRS + HIPEC	Cascales Campos, 2016 [24]			x		x		x					x	x	x
	Duzgun, 2019 [18]	x	x	x		x		x		x	x	x	x	x	x
	Webb, 2020 [21]		x			x		x	x	x			x		
	Siddhartan, 2020 [20]	x	x		x	x		x		x	x	x	x	x	x
	Lu, 2020 [22]	x	x		x	x		x				x	x	x	x
	Martin, 2020 [19]	x	x			x	x	x			x	x	x	x	x
	White, 2021 [23]	x	x			x	x	x	x	x	x	x	x	x	x
	Candiolo Cancer Institute	x	x	x		x	x	x			x	x	x	x	x
	Veneto Institute of Oncology	x		x		x	x				x	x	x	x	x
CRS	Marx, 2006 [25]			x		x	x			x		x	x	x	x
	Gerardi, 2008 [26]					x					x		x		
	Kalogera, 2013 [17]			x	x	x	x	x		x	x		x		x
	Myriokefalitak, 2016 [27]	x	x	x	x	x	x	x	x	x		x	x	x	x
	Modesitt, 2016 [28]	x			x	x							x	x	x
	Bisch, 2018 [29]	x	x	x	x		x		x	x		x	x	x	x
	Boitano, 2018 [31]	x		x	x	x		x	x	x		x	x	x	x
	Bergstrom, 2018 [33]	x		x	x	x	x	x	x	x		x	x	x	x
	Meyer 2018 [32]	x		x		x	x	x	x	x		x	x	x	x
	Mendivil, 2018 [38]	x		x	x	x	x	x					x	x	x
	Agarwal, 2019 [30]	x	x	x	x	x	x	x	x	x		x	x	x	x
	Sanchez- Iglesias, 2020 [35]	x	x	x	x	x	x	x	x	x		x	x		x
	Ferrari, 2020 [37]	x	x	x	x	x	x	x	x	x		x	x	x	x
	Bernard, 2020 [34]	x	x	x	x		x	x	x	x		x	x	x	x
	Kay, 2020 [39]		x	x	x	x		x	x				x	x	x
	Tankou, 2021 [36]	x		x	x	x		x					x	x	x
	Reuter, 2021 [40]	x		x	x	x	x					x			x
	Candiolo Cancer Institute	x	x	x		x	x	x			x	x	x	x	x
	Veneto Institute of Oncology	x		x		x	x				x	x	x	x	x

## Data Availability

Data and code for analysis are available upon request to the authors.

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
