# Peer review of "Enhanced Recovery after Surgery (ERAS) Program for Patients with Peritoneal Surface Malignancies Undergoing Cytoreductive Surgery with or without HIPEC: A Systematic Review and a Meta-Analysis"

_cancers, 2023, doi:10.3390/cancers15030570_

Round 1
Reviewer 1 Report
The authors perform a meta-analysis of ERAS program in patients undergoing CRS +/- HIPEC. This is the first meta-analysis concerning the application of the ERAS in such patients confirming its the feasibility and safety. They also conclude that implementation of an ERAS protocol may reduce LOS, postoperative complications after CRS +/- HIPEC compared to conventional recovery.
The study is well devised, and conclusions are sound.
There are scattered typographical and grammatical errors that can be easily fixed. I will point out a couple: "annissectomy" (line 185) and "IBMA showed es showed..." (line 214).
I congratulate the authors on performing this meta-anaylsis.
Author Response
I thank the reviewer for the appreciations and comments. Grammatical errors and typos have been corrected.
Reviewer 2 Report
The manuscript is written in a clear manner. However, I have a few concerns:
1. The introduction lacks a precise description of what ERAS is all about. Authors should redraft this section in order to provide better background for the reader.
2. Why were case reports not included in the selection of studies?
3. The graphical presentation of the results is very opaque, which makes it difficult for the reader to assimilate
Author Response
I thank the reviewer for the comments and suggestions.
- The introduction has been implemented with a detailed description of the ERAS program.
- given that no case reports of interest for the planned analysis were identified, we would not have included them given the already heterogeneous compliance of the various studies with the ERAS program; including case reports would have done nothing but increase this heterogeneity, without significantly enlarging the sample.